# Serum Phosphorus and Calcium as Biomarkers of Disease Status in Acromegaly

**DOI:** 10.3390/biomedicines11123278

**Published:** 2023-12-11

**Authors:** Nadia Sawicka-Gutaj, Aleksandra Derwich-Rudowicz, Aleksandra Biczysko, Solomiya Turchyn, Paulina Ziółkowska, Katarzyna Ziemnicka, Paweł Gut, Kacper Nijakowski, Marek Ruchała

**Affiliations:** 1Department of Endocrinology, Metabolism and Internal Medicine, Poznan University of Medical Sciences, 60-355 Poznan, Poland; 2Department of Conservative Dentistry and Endodontics, Poznan University of Medical Sciences, 60-812 Poznan, Poland

**Keywords:** growth hormone, IGF-1, acromegaly

## Abstract

Acromegaly is a chronic disease caused by the hypersecretion of growth hormone (GH), leading to changes in the growth of visceral tissues and glucose impairment. Serum biomarkers that correlate with disease status are still unclear. This study aims to assess the potential of phosphorus and calcium as biomarkers in the clinical evaluation of patients with acromegaly and clarify their relationship with SAGIT and other common biomarkers. We retrospectively analyzed data from 306 medical records of patients with acromegaly hospitalized between 2015 and 2020. Factors such as patient biometrics, duration of disease, SAGIT score, tumor size, GH, insulin-like growth factor 1 (IGF-1), calcium, phosphorus, parathormone, and vitamin D were analyzed concerning current disease status (naïve, non-remission, remission). The results showed that serum phosphorus significantly correlated with IGF-1 and SAGIT scores for patients with active acromegaly. Specifically, the best predictor for the remission of acromegaly was a phosphorus level < 3.98 mg/dL and serum calcium levels < 9.88 mg/dL. Based on logistic regression, the higher the serum phosphorus level, the lower the odds of achieving remission (an increase in one unit leads to a decrease in the chance of about 80%). In conclusion, phosphorus and calcium can be effective biochemical markers for predicting disease status in acromegaly.

## 1. Introduction

Acromegaly is characterized by the chronic hypersecretion of growth hormone (GH) and insulin-like growth factor 1 (IGF-1), most commonly caused by a pituitary adenoma [1,2,3,4,5,6]. Typical symptoms include somatic changes, such as enlarged hands and feet; thickened skin; mandible growth; joint tenderness; and enlarged facial features involving the facial bones, lips, nose, and tongue [6]. Chronic exposure to GH and IGF-1 leads to growth in visceral and epithelial tissues and increased bone turnover [3,7]. Patients with acromegaly are also at a higher risk of cardiovascular diseases, malignancy, hypertrophic arthropathy, obstructive sleep apnea, and metabolic disorders [1,3,5,8]. These comorbidities significantly reduce the quality of life of patients suffering from acromegaly but also shorten life expectancy [9]. The main goal of treatment, including neurosurgical resection and pharmacotherapy with somatostatin analogs, is to normalize GH and IGF-1 levels, lower symptoms, reduce the risk of developing comorbidities, and prolong survival.

It has also been suggested that phosphorus and calcium balance is impaired in these patients [1,3,10,11]. These changes are a result of the complex interplay of GH, insulin-like growth factor-1, and regulatory hormones involved in mineral metabolism. GH and its direct and indirect effects through IGF-1 play an essential role in the growth, differentiation, and repair of bone and cartilage formation. Data on the risk of osteoporosis and changes in bone mineral density at first were conflicting, as some studies reported increased bone mineral density (BMD) in patients with acromegaly, some observed reduced BMD, and others did not find any difference in BMD between acromegaly patients and healthy subjects [12]. Now, there is evidence that patients with active acromegaly may suffer from secondary osteoporosis and have skeletal fragility with a high prevalence and incidence of vertebral fractures, even with normal BMD [12]. Hyperphosphatemia and hypercalcemia in acromegaly patients are mediated by IGF-1’s direct effect on the Na-Pi cotransporter in the proximal convoluted tubules of the kidneys [2,3,13]. The clinical status of acromegaly can be assessed using SAGIT, a comprehensive clinician-reported outcome instrument that analyzes five key features of the disease: signs and symptoms (S), associated comorbidities (A), GH levels (G), IGF-1 levels (I), and the tumor profile (T) [5,14]. Recent studies have demonstrated that serum phosphate levels correlate with SAGIT scores and disease status and may be a valuable predictor of acromegaly activity in patients [2,15]. The assessment of serum phosphorus (P) levels, in addition to SAGIT scoring, may be beneficial as a serum biochemical marker and predictor of remission status [2,13]. A few small-sample-size studies have supported the use of phosphorus as a biomarker; clear guidelines have yet to be established [2,3,13,16]. Calcium (Ca) levels also seem to be higher in patients with active acromegaly compared with those in controlled disease activity with medical or surgical treatment, probably because of increased intestinal calcium absorption, enhanced bone calcium release, and reduced urinary calcium elimination. Elevated GH levels induce the upregulation of intestinal calcium absorption through mechanisms such as the increased production of 1,25-dihydroxyvitamin D, the active form of vitamin D; consequently, this results in hypercalcemia. Increased bone turnover encompasses the concurrent processes of bone resorption (osteoclastic activity) and bone formation (osteoblastic activity). Heightened GH and IGF-1 levels stimulate osteoclasts, which accelerate the breakdown of bone tissue, releasing calcium and other mineral constituents into the bloodstream. Concurrent with increased osteoclastic resorption, there is a surge in osteoblastic activity. Osteoblasts experience enhanced differentiation and function under the influence of excessive GH, which leads to the production of new bone matrix. However, the balance between bone formation and resorption may be disrupted, often favoring resorption [13,17].

This study aims to assess the potential of the use of phosphorus and calcium as biomarkers in the clinical evaluation of patients with acromegaly and clarify their relationship with SAGIT and other common biomarkers (GH, IGF-1) in larger population sizes. It is essential to emphasize that previous studies in this field collectively constitute a limited dataset, none of which involved a patient cohort as large as ours [2,3,13,17]. Only one study addressed correlations with the SAGIT instrument [2]. We divided patients into three groups, naïve, non-remission, and remission, to accurately assess the changes in the various stages of the disease. To date, this is the largest real-life study investigating serum phosphorus and calcium as biomarkers of disease status in acromegaly.

## 2. Materials and Methods

### 2.1. Study Design

This study has a retrospective design. Medical records of all patients diagnosed with acromegaly hospitalized either for the first time or for routine control at the Department of Endocrinology, Metabolism, and Internal Medicine of Poznan University of Medical Sciences in Poland between January 2015 and December 2020 were analyzed. Clinical history, laboratory results (GH, IGF-1, P, Ca, PTH, vitamin D, creatinine), and endocrine tests were routinely recorded. Age, gender, diagnostic images, and BMI were also reviewed and documented. A detailed medical history was taken, including medications and supplements that could affect Ca/P concentrations. All parameters were measured in blood samples taken after overnight fasting. Acromegaly was diagnosed and treated according to the current guidelines [18,19]. All patients underwent dual-energy X-ray absorptiometry to exclude osteopenia or osteoporosis, diagnosed according to the current guidelines [20,21,22]. The exclusion criteria were impaired renal function (glomerular filtration rate less than 60 mL/min/1.73 m^2^), thyroid dysfunction, osteopenia or osteoporosis, calcium supplementation, and primary hyperparathyroidism. All the laboratory tests were performed using standard laboratory methods according to the manufacturers’ recommendations. Serum creatinine, calcium, and phosphorus were measured with the photometric method using a Cobas Integra 400+ biochemistry analyzer (Roche Diagnostics, Indianapolis, IN, USA). PTH was measured by using electrochemiluminescence (ECLIA) (Roche Diagnostics, Indianapolis, IN, USA). 25(OH)D levels were measured with the chemiluminescence (CMIA) method using an Alinity I Analyzer (Abbott Diagnostics, Illinois, IL, USA). The GH levels were assayed via electrochemiluminescence (ECLIA) using the Cobas e402 analyzer (Roche Diagnostics). IGF-1 levels were measured via the chemiluminescence (CMIA) method using a LIAISON Analyzer (DiaSorin Ltd., Saluggia, Italy). The serum phosphorus normal range was 2.7–4.5 mg/dL. The serum calcium normal range was 8.8–10.2 mg/dL. Nadir GH was measured after a 75 g glucose tolerance test. In diabetic patients and/or those treated with somatostatin analogs (SSA), serum GH levels were determined every 30 min, and the arithmetical mean was counted based on five measurements [18,19,23,24]. All patients underwent an MRI scan (or CT, in case of contraindications) of the pituitary gland to determine image characteristics: tumor size, intratumor hemorrhage, and invasion type. (MRI: Siemens Magnetom Avanto (serial number 26184 to 2017); Siemens Magnetom Skyra (serial number 145114, from 2017). CT: GE Healthcare Light Speed VCT (serial number 407067CN8, from 2017); Siemens Somatom Definition Edge (serial number 83543, from 2017)). The greatest diameter of the tumor was measured as the tumor size.

The patients were assigned to three groups, (A) naïve, (B) non-remission, and (C) remission, according to the criteria below:

(A) The diagnosis of naïve acromegaly was made when all the criteria were fulfilled:Elevation of IGF-1 above the age-adjusted upper normal range.Nadir GH above 1 ng/mL in 75 g oral glucose tolerance test (patients without diabetes) or random GH levels above 2.5 ng/mL (mean of five measurements repeated every 30 min—in patients with diabetes) [18,19,23,24].A pituitary gland tumor detected in magnetic resonance imaging (MRI) or computed tomography (CT) (in patients with contraindications for MRI).

(B) Acromegaly was considered active/uncontrolled (non-remission group) when IGF-1 and GH were elevated. Patients who could not be classified according to the above-mentioned criteria were excluded from the analysis. 

(C) Remission was defined as age-normalization of IGF-1 and normalization of GH (GH < 1 ng/mL in 75 g OGTT or random GH < 2.5 ng/mL in diabetic patients) [18,19,23,24].

The SAGIT instrument was completed using patients’ medical records. It reflects key components of acromegaly: signs and symptoms (S), associated comorbidities (A), GH levels (G), IGF-1 levels (I), and tumor features (T). Each of the above units was scored by a clinician: S(0–4), A(0–6), G(0–4), I(0–3), and T(0–5). The higher the score in each category and the total sum of points, the greater the advancement of a given factor and overall disease activity [5,14,25]. The Bioethical Committee of Poznan University of Medical Sciences approved this study and waived the requirement for informed consent given the study’s retrospective nature (Decision No. 633/22). All methods were performed following the relevant guidelines and regulations [26]. 

### 2.2. Statistical Evaluation

Descriptive statistics of quantitative variables were recorded in medians and quartile ranges (because of non-compliance with the normal distribution, they were assessed with the Shapiro–Wilk test). A continuous data comparison was performed with non-parametric tests, the Mann-Whitney test and the Kruskal–Wallis test, followed by the Dunn post hoc test. Qualitative variables were compared with Pearson’s Chi-square test. Also, Spearman’s correlation coefficients were determined between parameters of the calcium–phosphate metabolism and diagnostic parameters for acromegaly.

Receiver operating characteristic (ROC) analysis was performed to discriminate between the patients with remission and naïve and/or non-remission acromegaly, as well as naïve and non-remission patients, based on phosphorus and calcium serum levels. The potential cut-off point was proposed based on Youden’s index. In addition, a univariate and multivariate logistic regression model incorporating phosphorus and calcium serum levels as discriminating predictors for the remission of acromegaly was conducted with V-fold cross-validation.

The significance level was set at α = 0.05 for all analyses. The statistical analysis was performed with Statistica v.13.3 (StatSoft, Cracow, Poland).

## 3. Results

### 3.1. Clinical Characteristics

Data from 319 hospitalizations were analyzed: given our strict inclusion criteria, 13 patients were not eligible for the analysis, yielding a total of 306 medical records included in the final analysis. Anthropometric and clinical characteristics are presented in Table 1. Active acromegaly was diagnosed in 190 hospitalizations (62.1% of all hospitalizations): 45 patients diagnosed de novo (naïve group) and 145 hospitalizations of patients (non-remission group) who did not respond to the previous therapy (transsphenoidal surgery/pharmacotherapy/radiotherapy).

### 3.2. Laboratory Characteristics

Table 2 presents comparisons of the laboratory parameters. The highest phosphorus levels were found in patients diagnosed with naïve acromegaly, while patients in remission had significantly lower levels (Figure 1). Women and men did not differ in serum phosphorus levels (*p*-value = 0.301). Therefore, no separate gender analysis was performed. An analogous relationship was observed for calcium levels. In contrast, vitamin 25OHD and PTH concentrations were similar in all subgroups of patients.

Serum phosphorus and calcium concentrations significantly correlated with IGF-1 levels and SAGIT scores in patients with non-remission acromegaly (Table 3). Correlations between PTH and vitamin D and diagnostic parameters for acromegaly are available in Appendix A (Table A1). In the case of naïve acromegaly, only significant correlations for serum phosphorus levels were found. In addition, serum phosphorus could correlate with the determined elements of the SAGIT instrument, especially G and T, which were significantly related to all subgroups of acromegaly patients (Appendix A, Table A2).

The performed ROC analysis showed that serum phosphorus and calcium levels could predict the remission of acromegaly (Table 4). The highest phosphorus and calcium predictive values were observed to discriminate the remission group from the naïve patients—AUC = 0.755, cut-offs (Youden’s index) of 3.98 mg/dL and 0.763; cut-off (Youden’s index) of 9.88 mg/dL, respectively (Figure 2). Serum phosphorus levels below 3.98 mg/dL were predisposed to acromegaly remission with an accuracy of 74.8%. In turn, the naïve status among the active patients could be significantly predicted based on serum calcium levels; serum phosphorus concentrations demonstrated borderline significance in this case.

To better assess the ability of serum phosphorus to classify patients with acromegaly, logistic regression modeling was performed in comparison with calcium (Table 5). Indeed, the higher the serum phosphorus level, the lower the odds of achieving remission (an increase in one unit leads to a decrease in the chance of more than 80%). The same correlation was observed for serum calcium levels—the higher the serum calcium level, the lower the odds of achieving remission.

Considering serum calcium concentrations as a confounder, a multivariate logistic regression model was constructed to prognose the remission group from naïve acromegaly patients (Table 6). Both serum elements showed a good fitting for the discrimination of the indicated acromegaly subgroups.

## 4. Discussion

We aimed to investigate phosphorus and calcium as potential biomarkers in a large group of patients with acromegaly. We found a cut-off serum level for P to establish naïve/active patients vs. patients in remission. Serum P < 3.98 mg/dL predicted the remission of acromegaly, but it could not discriminate between the naïve and non-remission groups. Furthermore, the higher the serum P level, the lower the odds of achieving remission, as an increase in one unit led to a decrease in the chance by about 80%. Previous studies have shown that serum P is an important marker of morbidity and mortality, so its measurement can be indicative of complications [2,3]. Similar results for serum Ca in logistic regression support the conclusion that additional data about calcium levels may also be beneficial in the prediction of the remission of acromegaly. Our findings show that serum Ca levels ≤ 9.88 mg/dL predicted the remission of acromegaly, which supports similar findings by Piskinpasa et al., where disease status correlated with serum Ca levels [13,27]. 

In comparison with previous studies, our investigation involved a substantially larger sample size, comprising 306 medical records. To date, four studies have been published on this topic, with our research contributing significantly to understanding diagnostic markers and their role in the clinical management of acromegaly [2,3,13,17]. Unlike some earlier studies, our cohort encompassed three patient groups: naïve acromegaly, remission, and non-remission.

The first of the cited studies had a retrospective design and incorporated a total of 103 patients, providing valuable insights into post-surgical treatment outcomes [2]. However, it is noteworthy that the sample size was much smaller than our comprehensive study. One notable limitation of this study lies in the absence of a specific inclusion of a naïve acromegaly group.

The second cited study, a retrospective analysis of 51 acromegaly patients, confirmed results on a smaller scale without specifying a naïve acromegaly group [3]. In our study, we confirmed elevated phosphorus levels in the non-remission group and identified the highest phosphorus levels in patients with naïve acromegaly. Additionally, we confirmed a significant correlation between phosphorus levels and the SAGIT score. Notably, our study represents the largest real-life investigation to date of serum phosphorus and calcium as potential biomarkers for assessing disease status in acromegaly.

In the third cited study, which included 73 acromegaly patients, participants were categorized into active and controlled acromegaly groups [13]. However, this study did not explore correlations with the SAGIT instrument. In our study, we expanded the analysis to 306 medical records and included three groups: naïve, non-remission, and remission. We observed the highest phosphorus and calcium levels in patients with naïve acromegaly. Importantly, we confirmed a correlation between serum phosphorus levels and elements of the SAGIT instrument, especially G and T, suggesting that phosphorus could be an effective biomarker in acromegaly diagnosis.

Finally, in another study, encompassing 22 acromegaly patients and 22 patients with nonfunctioning pituitary adenomas, the focus was on patients assessed 3–6 months post-treatment [17]. In contrast to our study, the acromegaly active group did not include a division between naïve and non-remission diseases. In our research, we extended the follow-up period, particularly in the remission group, providing a more comprehensive perspective on treatment outcomes. We provide a detailed comparison of the above studies in Appendix (Table A3).

Moreover, we observed serum Ca > 9.89 mg/dL had the potency to discriminate between the naïve and non-remission groups. Building upon prior findings, our study not only confirmed elevated serum calcium and phosphorus levels in the non-remission group but also revealed the highest calcium and phosphorus levels in patients diagnosed with naïve acromegaly. Furthermore, our investigation established a significant correlation between phosphorus levels and the SAGIT score, contributing to a more nuanced understanding of the biomarker’s relevance in acromegaly.

Serum Ca and P increase in patients with acromegaly because of increased bone turnover, characterized by a combination of bone formation and resorption with bone absorption slightly dominant and serum calcium balanced. Markers of bone formation and resorption, osteocalcin, B-ALP, n-terminal telopeptide, and CTX, generally increase in active disease and tend to normalize after remission [17,28]. The other cause is the downstream effect of the GH/IGF-1 axis via renal tubular reabsorption [2,16,29]. Hypercalcemia and hyperphosphatemia in these patients are usually due to the downstream effects of IGF-1 on the Na-Pi cotransporter in the proximal convoluted tubules of the kidneys [2,3,13]. Patients with acromegaly frequently have mild hyperphosphatemia because of increased renal P reabsorption in the proximal tubules. A higher positive correlation between P levels and SAGIT scores was observed both in preoperative acromegaly patients and 1-year postoperative non-remission patients than the correlations between other factors such as GH, OGTT-GH, IGF-1, IGBP-3, IGF-1/IGBP-3, and SAGIT scores, but there was no correlation between hyperphosphatemia and remission rates [2]. Baseline serum P levels decreased significantly after treatment in all the patients, and the mean baseline serum P levels were higher in patients in non-remission [3]. Therefore, P represents the target organ response to GH hypersecretion, closely reflecting the disease status, making Ca and P useful potential biomarkers. While the literature agrees that the highest level of serum P is in patients with active acromegaly, there is no consensus regarding the correlation between disease status and P levels [2,3,13,16]. In our study, serum P levels correlated with disease status, where the highest levels were found in patients with naïve acromegaly and the lowest levels in patients with remission. Similar findings were described by Povaliaeva et al., where an increase in the serum P level was the most prevalent feature in the biochemical examination and correlated with the activity of the disease [30]. This discrepancy between previous studies can be explained by the power of the study, as both previous studies had smaller sample sizes [2,3,13]. Women and men did not differ in serum phosphorus levels. The aforementioned studies also did not demonstrate an association between gender and calcium or phosphorus levels; disease status emerged as the principal factor exhibiting the strongest correlation with alterations in Ca and P levels. 

Serum P levels additionally correlated with SAGIT scores and IGF-1 in the naïve and non-remission groups. This finding is supported by the literature, where a statistically significant correlation between preoperative and postoperative levels and SAGIT scores has been established [2]. Our recent study on SAGIT’s potential utility in the clinical assessment of patients with acromegaly revealed positive correlations between SAGIT scores and concentrations of Ca, P, HbA1C levels, and tumor invasiveness at the time of diagnosis [25]. However, parameters such as age, vitamin D concentration, and time from diagnosis showed an inverse relationship with the SAGIT score [25]. The correlation of serum P with IGF-1 levels helps to resolve a dispute between smaller studies that did not find significant correlation and supports findings by Xie et al. and Piskinpasa et al. [2,3,13]. 

As already mentioned, hyperphosphatemia can be an important marker of morbidity and mortality in acromegaly patients. Initially, phosphorus levels were considered an indicator of bone mineral disease and a predictor of vertebral fractures. However, some recent small-scale studies have demonstrated that phosphorus does not correlate with bone mineral density (BMD) preoperatively [29]. Calayutad et al. found that most acromegaly patients have a normal BMD but an impaired trabecular bone score (TBS) [31]. TBS is a more accurate determinant of bone microarchitecture and bone strength and is significantly reduced in acromegaly patients, improving postoperatively [31]. A meta-analysis that included 1935 acromegaly patients showed both higher bone formation and bone resorption as compared with healthy controls, without significant differences in lumbar spine BMD. Markers of bone resorption were more elevated compared with markers of bone formation and were significantly higher in patients with active disease. On the other hand, bone mineral density at the femoral neck was higher in acromegaly patients compared with the controls. Low BMD is not a common clinical feature in patients with acromegaly, and gonadal status appears to be the most influential factor. The reason for this might be underestimated measurements in dual-energy X-ray absorptiometry (DXA) due to bone enlargements caused by GH excess. Vertebral fractures occur more frequently in males than females, especially in close relationships with untreated hypogonadism and active acromegaly [12].

Increased serum P stimulates FGF-23 and PTH, which play a role in cardiovascular disease mortality and hypertension, especially in patients with chronic kidney disease. Phosphorus does this through increased endothelial dysfunction and valvular smooth muscle calcification [32,33,34,35,36,37].

While our study is the largest to date—with 306 medical records analyzed and introducing the novelty of dividing patients into three groups, naive, without remission, and with remission—it had certain limitations. We applied strict inclusion criteria and excluded all patients with impaired renal function, thyroid dysfunction, osteopenia or osteoporosis, calcium supplementation, and primary hyperparathyroidism to avoid potential bias. Similar exclusion criteria have been used in previous studies. Nevertheless, in using a retrospective study design, it is possible that not all confounding factors were considered. Bone mineral metabolism can change because of dietary changes, physical activity, and vitamin D deficiency. Disproportion between genders could have influenced the calcium and phosphorus results. This study only reflected the patient population in one facility in Poznan, Poland. The cut-off of GH in 75 g OGTT was <1 ng/mL since it is a method used in our laboratory. Further research should also assess the relationships of serum P to TBS and BMD values to determine any causal relationships.

## 5. Conclusions

In conclusion, phosphorus and calcium can be effective biochemical markers for predicting disease status in acromegaly. The best predictor of the remission of acromegaly was a phosphorus level < 3.98 mg/dL and serum calcium levels < 9.88 mg/dL. Our paper establishes serum phosphorus and calcium as markers of naïve acromegaly and supports phosphorus as a biochemical marker for remission in acromegaly. 

## Figures and Tables

**Figure 1 biomedicines-11-03278-f001:**
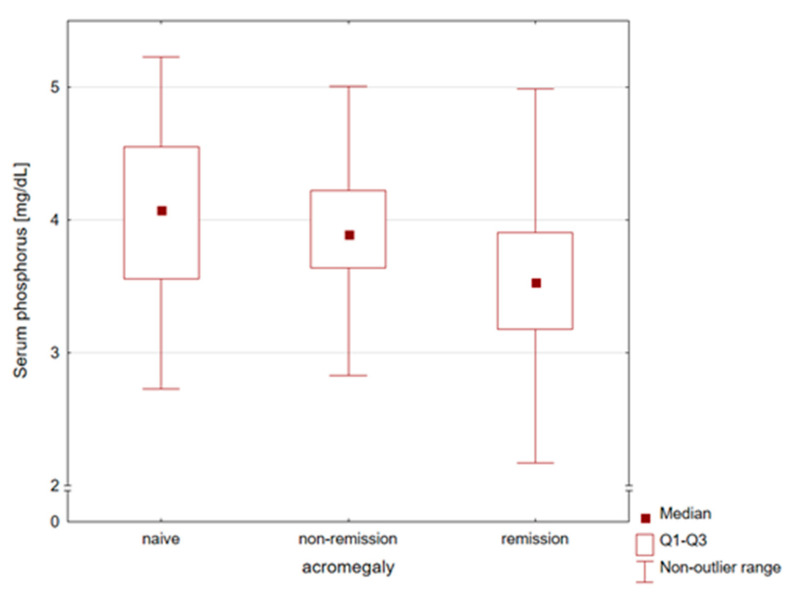
Comparison of serum phosphorus concentrations in naïve, non-remission, and remission patients with acromegaly. The highest phosphorus levels were found in patients diagnosed with naïve acromegaly, while patients in remission had significantly lower levels.

**Figure 2 biomedicines-11-03278-f002:**
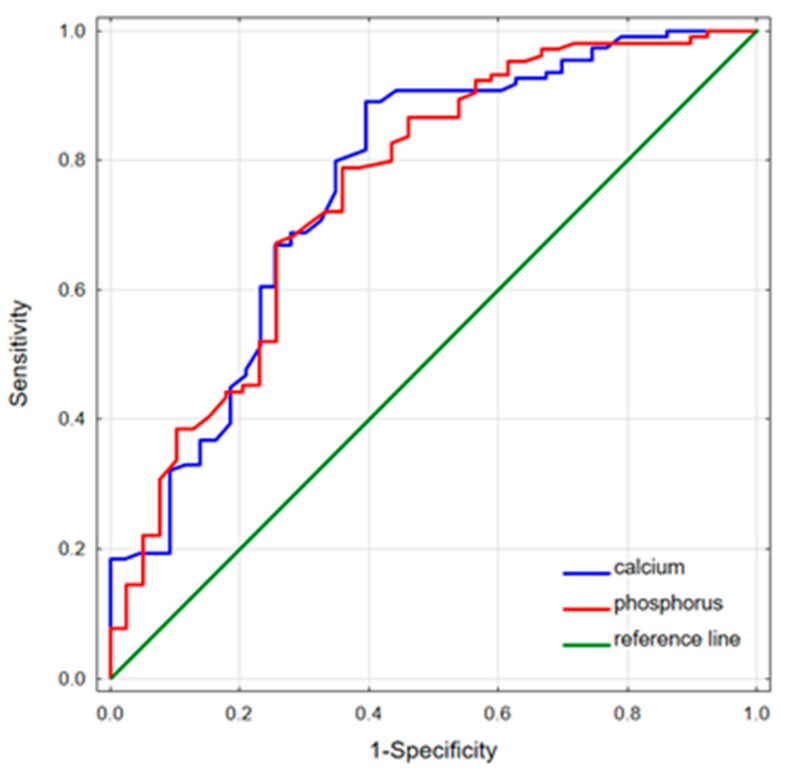
ROC curves for serum calcium and phosphorus discriminating the remission group from the naïve patients. Phosphorus predictive values: AUC = 0.755; cut-off (Youden’s Index), 3.98 mg/dL (sensitivity 0.788, specificity 0.641); calcium predictive values: AUC = 0.763; cut-off (Youden’s Index), 9.88 mg/dL (sensitivity 0.890, specificity 0.605).

**Table 1 biomedicines-11-03278-t001:** Anthropometric and clinical characteristics (continuous data presented as medians with quartile ranges; *p*-values for the Kruskal–Wallis test followed by the Dunn post hoc test, except for the variables gender—Pearson’s Chi-square test—and duration time—the Mann–Whitney test).

Parameter	Naïve (A)	Non-Remission (B)	Remission (C)	*p*-Values	*p*-Values Post Hoc		
	N = 45	N = 145	N = 116		A vs. B	A vs. C	B vs. C
Gender (female)	22	95	81	0.043 *	n/a	n/a	*n*/a
Age (years)	(*n* = 45) 54 (39–62)	(n = 145) 56 (45–65)	(n = 116) 59.5 (47–64.5)	0.035 *	0.100	0.030 *	>0.999
Body mass (kg)	(n = 42) 82 (74–95)	(n = 135) 85 (70–96)	(n = 107) 81 (71–94)	0.890	>0.999	>0.999	>0.999
BMI (kg/m^2^)	(n = 42) 28.2 (24.7–31.8)	(n = 135) 28.1 (25.5–32.6)	(n = 106) 29 (25.7–33.6)	0.759	>0.999	>0.999	>0.999
Duration time from diagnosis (months)	n/a	(n = 140) 48 (21–99.5)	(n = 109) 132 (48–186)	<0.001 *	n/a	n/a	n/a
Tumor size (mm)	(n = 45) 11 (9–14)	(n = 137) 11 (0–17.25)	(n = 113) 0 (0–8)	<0.001 *	0.704	<0.001 *	<0.001 *
SAGIT:	(n = 45) 11 (9–14)	(n = 145) 8 (5–11)	(n = 115) 3 (3–5)	<0.001 *	0.022*	<0.001 *	<0.001 *
S	1 (0–2)	0 (0–1)	0 (0–1)	0.009 *	0.021 *	0.031 *	>0.999
A	2 (1–2)	2 (1–3)	2 (1–3)	0.015 *	0.073	>0.999	0.066
G	4 (3–4)	2 (2–3)	0 (0–1)	<0.001 *	0.006 *	<0.001 *	<0.001 *
I	3 (2–3)	1 (0–3)	0 (0–0)	<0.001 *	0.003 *	<0.001 *	<0.001 *
T	2 (1–4)	2 (0–4)	0 (0–1)	<0.001 *	0.062	<0.001 *	<0.001 *

* statistically significant, n/a—not applicable.

**Table 2 biomedicines-11-03278-t002:** Comparisons of laboratory parameters (continuous data presented as medians with quartile ranges; *p*-values for the Kruskal–Wallis test followed by the Dunn post hoc test).

Parameter	Naïve (A)	Non-Remission (B)	Remission (C)	*p*-Values	*p*-Values Post Hoc		
	N = 45	N = 145	N = 116		A vs. B	A vs. C	B vs. C
GH random (ng/mL)	(n = 16) 10.15 (4.60–16.58)	(n = 67) 4.24 (1.91–8.84)	(n = 32) 0.72 (0.44–1.34)	<0.001 *	0.174	<0.001 *	<0.001 *
nadir GH or GH-in OGTT (ng/mL)	(n = 28) 8.09 (3.33–17.60)	(n = 77) 2.35 (1.54–4.33)	(n = 83) 0.62 (0.28–0.99)	<0.001 *	0.006 *	<0.001 *	<0.001 *
IGF-1 (ng/mL)	(n = 43) 702 (553–943)	(n = 140) 342 (219.5–568.5)	(n = 115) 167 (116–214)	<0.001 *	<0.001 *	<0.001 *	<0.001 *
Calcium (mg/dL)	(n = 43) 9.93 (9.59–10.23)	(n = 135) 9.69 (9.48–9.97)	(n = 109) 9.53 (9.33–9.70)	<0.001 *	0.051	<0.001 *	<0.001 *
Phosphorus (mg/dL)	(n = 39) 4.08 (3.56–4.55)	(n = 120) 3.90 (3.64–4.22)	(n = 104) 3.53 (3.18–3.91)	<0.001 *	0.255	<0.001 *	<0.001 *
PTH (pg/mL)	(n = 32) 35.88 (26.33–48.03)	(n = 85) 41.61 (33.35–55.61)	(n = 70) 42.35 (31.37–54.64)	0.135	0.166	0.234	>0.999
Vitamin D (ng/mL)	(n = 23) 21 (14–33)	(n = 69) 23 (16–32)	(n = 60) 25 (16–36)	0.506	>0.999	>0.999	0.989

Legend: GH, growth hormone; OGTT, oral glucose tolerance test; IGF-1, insulin-like growth factor 1; PTH, parathyroid hormone; *, significant differences for *p*-value < 0.05.

**Table 3 biomedicines-11-03278-t003:** Correlations between calcium and phosphorus and diagnostic parameters for acromegaly (Spearman’s rank correlation coefficients).

Parameter	Group	SAGIT		GH (ng/mL)		IGF-1 (ng/mL)	
		Rs	*p*-Value	Rs	*p*-Value	Rs	*p*-Value
Calcium (mg/dL)	Naïve	0.269	0.081	0.115	0.470	0.236	0.138
	Non-remission	0.266	0.002 *	0.118	0.175	0.353	<0.001 *
	Remission	0.101	0.298	−0.201	0.037 *	0.079	0.415
Phosphorus (mg/dL)	Naïve	0.464	0.003 *	0.622	<0.001 *	0.533	0.001 *
	Non-remission	0.373	<0.001 *	0.469	<0.001 *	0.304	0.001 *
	Remission	0.086	0.388	0.038	0.702	0.046	0.644

Legend: GH, growth hormone; IGF-1, insulin-like growth factor 1; *, significant correlations for *p*-values < 0.05.

**Table 4 biomedicines-11-03278-t004:** Results of ROC analysis for serum calcium and phosphorus concentrations with proposed cut-offs based on Youden’s index (and sensitivity, specificity, and accuracy for these points).

Parameter	Positive Group	Negative Group	AUC	SE	*p*-Value	Cut-Off (Youden’s Index)	Sensitivity	Specificity	Accuracy
Calcium (mg/dL)	Remission	Naïve	0.763	0.046	<0.001 *	<9.88	0.890	0.605	0.809
	Remission	Non-remission	0.659	0.035	<0.001 *	<9.61	0.670	0.622	0.643
	Remission	Naïve + non-remission	0.684	0.032	<0.001 *	<9.61	0.670	0.652	0.659
	Naïve	Non-remission	0.632	0.051	0.009 *	>9.89	0.605	0.674	0.657
Phosphorus (mg/dL)	Remission	Naïve	0.755	0.048	<0.001 *	<3.98	0.788	0.641	0.748
	Remission	Non-remission	0.683	0.036	<0.001 *	<3.69	0.606	0.717	0.665
	Remission	Naïve + non-remission	0.700	0.033	<0.001 *	<3.69	0.606	0.723	0.677
	Naïve	Non-remission	0.604	0.056	0.060	>3.99	0.641	0.600	0.610

Legend: AUC, area under curve; SE, standard error; * statistically significant.

**Table 5 biomedicines-11-03278-t005:** Results of univariate logistic regression for serum calcium and phosphorus concentrations.

Parameter	Positive Group	Negative Group	Beta	SE	*p*-Value	OR	−95% CI	+95% CI
Calcium (mg/dL)	Remission	Naïve	−2.946	0.614	<0.001 *	0.053	0.016	0.175
	Remission	Non-remission	−1.576	0.396	<0.001 *	0.207	0.095	0.450
	Remission	Naïve + non-remission	−1.832	0.375	<0.001 *	0.160	0.077	0.334
	Naïve	Non-remission	1.038	0.420	0.013 *	2.823	1.240	6.426
Phosphorus (mg/dL)	Remission	Naïve	−1.759	0.394	<0.001 *	0.172	0.080	0.373
	Remission	Non-remission	−1.216	0.273	<0.001 *	0.296	0.174	0.506
	Remission	Naïve + non-remission	−1.310	0.253	<0.001 *	0.270	0.164	0.443
	Naïve	Non-remission	0.641	0.338	0.058	1.899	0.980	3.680

* statistically significant.

**Table 6 biomedicines-11-03278-t006:** Results of multivariate logistic regression for modeling of remission group (vs. naïve group) with V-fold cross-validation (reported AUC for training and testing curves).

Parameter	Beta	SE	*p*-Value	OR	−95% CI	+95% CI	AUC Training	SE	AUC Testing	SE
Intercept	35.953	7.321	<0.001 *				0.836	0.039	0.817	0.041
Calcium (mg/dL)	−3.033	0.730	<0.001 *	0.048	0.012	0.202				
Phosphorus (mg/dL)	−1.450	0.422	0.001 *	0.235	0.103	0.537				

Legend: SE, standard error; OR, odds ratio; CI, confidence interval; AUC, area under curve; * statistically significant.

## Data Availability

The data that support the findings of this study are available from the corresponding author upon reasonable request.

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
