# Peer review of "Serum Phosphorus and Calcium as Biomarkers of Disease Status in Acromegaly"

_biomedicines, 2023, doi:10.3390/biomedicines11123278_

Round 1
Reviewer 1 Report (Previous Reviewer 1)
Comments and Suggestions for Authors
The authors responded to all my comments.
Author Response
Dear Reviewer,
We wanted to express our sincere gratitude for taking the time to review our paper. Your feedback is highly valuable to us, and we appreciate the insights you shared.
Reviewer 2 Report (Previous Reviewer 2)
Comments and Suggestions for Authors
The authors have modified the manuscript to a better level, however, there are a few points :
1- details for methods of measurement of IGF, GH , Ca, Vit D...kit??
2- name and company of CT and MRI apparatus.....model?
3- put table 2 entirely in separate page, avoid splitting the table?
Comments on the Quality of English Languageminor punctuation errors like m2 instead of m2
Author Response
Dear Reviewer,
Thank you for your valuable comments. We provide point-by-point response:
1) details for methods of measurement of IGF, GH , Ca, Vit D...kit??
Thank you for that remark. We added the following details:
“All the laboratory tests were performed using standard laboratory methods according to the manufacturer's recommendations. Serum creatinine, calcium and phosphorus were measured by the photometric method using a Cobas Integra 400+ biochemistry analyzer (Roche Diagnostics). PTH was measured by using electrochemiluminescence (ECLIA) (Roche Diagnostics). 25(OH)D levels were measured by chemiluminescence (CMIA) method using a Alinity I Analyzer (Abbott Diagnostics). The GH levels were assayed by electrochemiluminescence (ECLIA) using Cobas e402 analyzer (Roche Diagnostics). IGF-1 levels were measured by chemiluminescence (CMIA) method using a LIAISON Analyzer (DiaSorin Ltd).”
2) name and company of CT and MRI apparatus.....model?
We specified the name and company of CT and MRI apparatus model.
“All patients underwent an MRI scan (or CT, in case of contraindications) of the pituitary gland to determine image characteristics: tumor size, intratumor hemorrhage, and invasion type [MRI: Siemens Magnetom Avanto (serial number 26184 to 2017); Siemens Magnetom Skyra (serial number 145114; from 2017). CT: GE Healthcare Light Speed VCT (serial number 407067CN8, to 2017); Siemens Somatom Definition Edge (serial number 83543, from 2017)].”
3) put table 2 entirely in separate page, avoid splitting the table?
Thank you for your comment, we placed Table 2 on the next page to avoid any splitting. We also checked other tables and figures. We understand the importance of presenting the information in a clear and organized manner. We also corrected the spelling.
Reviewer 3 Report (Previous Reviewer 3)
Comments and Suggestions for Authors
Thank you for incorporating the changes. I think that the manuscript profits from these changes.
Author Response
Dear Reviewer,
Thank you for all your comments and suggestions. They have been incredibly valuable, and we are grateful for your insightful feedback.
This manuscript is a resubmission of an earlier submission. The following is a list of the peer review reports and author responses from that submission.
Round 1
Reviewer 1 Report
Comments and Suggestions for Authors
This study lacks novelty. The idea of this study is repeated in other studies as presented here:
1) Bone biochemical markers in acromegaly: An association with disease activity and gonadal status.
2) Calcium and Bone Turnover Markers in Acromegaly: A Prospective, Controlled Study
3) Utility of baseline serum phosphorus levels for predicting remission in acromegaly patients.
4) Serum phosphate: Does it more closely reflect the true state of acromegaly?
5) Calcium and Bone Turnover Markers in Acromegaly: A Prospective, Controlled Study
Comments on the Quality of English Language
Moderate editing of English language required
Author Response
Dear Reviewer,
Thank you for taking the time to review our manuscript. Below, we provide point-by-point response to your comments.
Comments: This study lacks novelty. The idea of this study is repeated in other studies as presented here:
- Bone biochemical markers in acromegaly: An association with disease activity and gonadal status
- Calcium and Bone Turnover Markers in Acromegaly: A Prospective, Controlled Study
- Utility of baseline serum phosphorus levels for predicting remission in acromegaly patients
- Serum phosphate: Does it more closely reflect the true state of acromegaly?
- Calcium and Bone Turnover Markers in Acromegaly: A Prospective, Controlled Study
Response: Thank you for your comment. We discussed the results of all the mentioned studies in our paper. It is essential to emphasize that these studies collectively constitute a limited dataset, encompassing only four investigations, none of which involved a patient cohort as large as ours. We would like to point out the differences and novelty in our study, which we also highlighted in the discussion:
1. Bone biochemical markers in acromegaly: An association with disease activity and gonadal status.
This study included 73 patients with acromegaly and 64 healthy controls. Patients with acromegaly were divided into active/controlled acromegaly groups. The study did not address correlations with SAGIT instrument. In contrast, in our study we analyzed 306 medical records, divided into three groups: naïve, non-remission and remission. We observed the highest phosphorus and calcium levels in patients diagnosed with naïve acromegaly. We also found correlation between serum phosphorus and determined elements of the SAGIT instrument, especially G and T, which suggest that phosphorus could be an effective biomarker.
2. Calcium and Bone Turnover Markers in Acromegaly: A Prospective, Controlled Study
This study included 22 patients with acromegaly referred for surgical or medical therapy and 22 with nonfunctioning pituitary adenomas referred for surgery as control. In contrary to our study, only patients with naïve acromegaly were included. Patients were assessed 3 to 6 months after treatment. In our study we had a longer follow-up in remission group.
3. Utility of baseline serum phosphorus levels for predicting remission in acromegaly patients
This retrospective study included 51 patients with acromegaly. Patients with acromegaly were divided into active/controlled, there was no naïve group. We confirmed observed results on a larger group (51 vs 306) and we corelated them with SAGIT.
4. Serum phosphate: Does it more closely reflect the true state of acromegaly?
The research design was also retrospective and comprised 103 patients. Nonetheless, the sample size was still nearly three times smaller than our study, and all patients were evaluated post-surgical treatment, with no inclusion of a naïve acromegaly group. We confirmed higher serum phosphorus in the non-remission group and found the highest phosphorus in patients diagnosed with naïve acromegaly. We also confirmed correlation between the P level and SAGIT score. To the date this is the largest real-life study investigating serum phosphorus and calcium as biomarkers of disease status in acromegaly.
5. Calcium and Bone Turnover Markers in Acromegaly: A Prospective, Controlled Study
We discussed that paper in subpoint 2.
Reviewer 2 Report
Comments and Suggestions for Authors
The manuscript is generally well-written but there are some points that need some improvements:
1- please avoid using abbreviations in the abstract, at lease define when first used
2- clarify in figure legends the way the results are expressed
3- there is duplication in expressing ROC curve once as a table and the other as a figure, I guess the fig is enough ( Fig 2= table 5). However, they are not clearly stated in the written results
4- have the authors studied any gender differences among studied groups? or at least add to the discussion if previous studies pointed to that
5- conclusion should summarize the basis for the stated conclusion
Comments on the Quality of English Language
minor punctuation errors
Author Response
Dear Reviewer,
Thank you for taking the time to review our manuscript. Below, we provide point-by-point response to your comments.
The manuscript is generally well-written but there are some points that need some improvements:
Comments 1- please avoid using abbreviations in the abstract, at lease define when first used
Response: Thank you for your comment. We corrected the abstract.
Comments 2- clarify in figure legends the way the results are expressed
Response: We changed figures descriptions (lines 188-190; 224-228)
Comments 3- there is duplication in expressing ROC curve once as a table and the other as a figure, I guess the fig is enough ( Fig 2= table 5). However, they are not clearly stated in the written results
Response: We changed Figure 2 to be more precise. Nevertheless, we left the Table 5 (after corrections- Table 4) as it provides more detailed data on results of ROC analysis with proposed cut-offs based on Youden’s index (and sensitivity, specificity, and accuracy for these points).
Comments 4- have the authors studied any gender differences among studied groups? or at least add to the discussion if previous studies pointed to that
Response: Thank you for that remark. Women and men did not differ in serum phosphorus levels (p-value=0.301). Therefore, no separate gender analysis was performed. Logistic regression modelling was also tested, and although gender could be a significant predictor in the univariate model, it was outside the multivariate model based on progressive stepwise regression. The discussed studies also did not demonstrate an association between gender and calcium or phosphorus levels. We added this information in the discussion.
Comments 5- conclusion should summarize the basis for the stated conclusion
Response: We changed conclusions to: “In conclusion, phosphorus and calcium can be effective biochemical markers for predicting disease status in acromegaly. The best predictor for remission of acromegaly was a phosphorus level<3.98 mg/dL and serum calcium levels<9.88 mg/dL. Our paper establishes serum phosphorus and calcium as markers of naïve acromegaly and supports phosphorus as a biochemical marker for remission in acromegaly.”
Reviewer 3 Report
Comments and Suggestions for Authors
The manuscript "Serum phosphorus and calcium as biomarkers of disease status in acromegaly" by Sawicka-Gutaj et al. presents an interesting retrospective analysis of a fairly large set of patients sub-divided into treatment naïve patients, biochemically active patients (despite treatment), and patients in remission investigating serum calcium and phosphorus levels as well as the SAGIT score. The main findings are a significant correlation of serum phosphorus/calcium with disease status/IGF-I as well as SAGIT score. The authors derive from their data a phosphorus cut-off discerning remission from active disease. The findings support the claim that serum phosphorus and calcium may be effective biomarkers of disease status in acromegaly.
I already had the pleasure of reviewing a paper of this research group a few months ago, which was published in February and is also cited in this manuscript. Again, the manuscript is mostly well written and very interesting. I enjoyed reading it very much.
However, there are a few issues I'd like to point out that should be addressed.
* line 80f.: ref #19 cites Giustina 2000 which is quite antique, as is the cut-off of 1 ng/ml for GH after oral glucose load for diagnosing acromegaly; that needs at least to be discussed in the limitations
* line 117f.: "due to exclusion criteria" - as stated in line 85f. the "exclusion criteria" are the non-applicability of inclusion criteria; so this should be rephrased, e.g.: "due to our strict inclusion criteria, 13 patients were not eligible for the analysis, yielding a total of 306 medical records included in the final analysis"
* table 3 is a lot to take in; maybe it's an option to leave out PTH and Vitamin D (?)
* table 4 seems not very meaningful: what good is a highly significant and high correlation of phosphorus with the SAGIT-G within the confines of treatment naïve acromegaly patients?
* figure 2 contains 2 ROC curves labeled "phosphorus" - I suppose one of them is calcium?
* where do the AUC values from figure 2 come from? they are not to be found anywhere (text, tables)
* line 171f.: "The exact relationship occured for serum calcium levels." What does that mean?
* I don't understand table 6 at all - where's phosphorus?
* line 175: it should be "confounder"
Comments on the Quality of English LanguageThe language is mostly fine, there are a few odd sentences though.
Author Response
Dear Reviewer,
Thank you for taking the time to review our manuscript. Below, we provide point-by-point response to your comments.
Comment 1* line 80f.: ref #19 cites Giustina 2000 which is quite antique, as is the cut-off of 1 ng/ml for GH after oral glucose load for diagnosing acromegaly; that needs at least to be discussed in the limitations
Response: Thank you for your remark, we mentioned that in limitations: “The cut-off of GH in 75g OGTT was < 1 ng/mL, since it was a method used in our laboratory.” (line 371)
Comment 2* line 117f.: "due to exclusion criteria" - as stated in line 85f. the "exclusion criteria" are the non-applicability of inclusion criteria; so this should be rephrased, e.g.: "due to our strict inclusion criteria, 13 patients were not eligible for the analysis, yielding a total of 306 medical records included in the final analysis"
Response: Thank you for your comment, we changed this sentence according to your instructions (line 157).
Comment 3* table 3 is a lot to take in; maybe it's an option to leave out PTH and Vitamin D (?)
Response: Thank you for your comment. We moved PTH and Vitamin D results to supplementary materials- Appendix (Table A1).
Comment 4* table 4 seems not very meaningful: what good is a highly significant and high correlation of phosphorus with the SAGIT-G within the confines of treatment naïve acromegaly patients
Response: The table shows that serum phosphorus could correlate with determined elements of the SAGIT instrument, especially G and T, which were significantly related to all subgroups of acromegaly patients and could be an effective biomarker. We moved that table to supplementary materials- Appendix (Table A2).
Comment 5* figure 2 contains 2 ROC curves labeled "phosphorus" - I suppose one of them is calcium?
Response: We changed Figure 2 to be more precise (line 214).
Comment 6* where do the AUC values from figure 2 come from? they are not to be found anywhere (text, tables)
Response: We changed Figure 2 to be more precise (line 214).
Comment 7* line 171f.: "The exact relationship occured for serum calcium levels." What does that mean?
Response: We rewrote this sentence – “The same correlation was observed for serum calcium levels - the higher the serum calcium level, the lower the odds of achieving remission.” (line 230)
Comment 8* I don't understand table 6 at all - where's phosphorus?
Response: We are very sorry for that. Now the table is corrected (Table 5, line 235)
Comment 9* line 175: it should be "confounder"
Response: Thank you for your remark. We corrected the spelling (line 274).
Reviewer 4 Report
Comments and Suggestions for Authors
This is a retrospective study aiming to assess the potential for the use of phosphorus and calcium as biomarkers in the clinical evaluation of patients with acromegaly.
The idea is not new and the work is potentially valuable. However, there are so many potential biases/confounding factors that have not been taken into account that, in my opinion, the results are not reliable. Examples: Ca/vitamin D supplements, osteoporosis diagnosis and/or treatments that could alter the serum calcium level (the study sample consists of mainly women, mostly postmenopausal), calciuria, renal function etc. All these factors should be taken into account and controlled for in order to adequately interpret the results
Comments on the Quality of English LanguageAdequate
Author Response
Dear Reviewer,
Thank you for taking the time to review our manuscript. Below, we provide point-by-point response to your comments.
Comment: The idea is not new and the work is potentially valuable. However, there are so many potential biases/confounding factors that have not been taken into account that, in my opinion, the results are not reliable. Examples: Ca/vitamin D supplements, osteoporosis diagnosis and/or treatments that could alter the serum calcium level (the study sample consists of mainly women, mostly postmenopausal), calciuria, renal function etc. All these factors should be taken into account and controlled for in order to adequately interpret the results
Response: According to the applied criteria all patients with impaired renal function (glomerular filtration rate less than 60 ml/min/1.73m2), thyroid dysfunction, osteopenia or osteoporosis, calcium supplementation, and primary hyperparathyroidism were excluded from the study. To the date this is the biggest “real-life” study investigating serum phosphorus and calcium as biomarkers of disease status in acromegaly.
Round 2
Reviewer 1 Report
Comments and Suggestions for Authors
It is important to acknowledge the limitations of previous studies in the same field and explain how the current study addresses them. This provides a clear rationale for the research question and the aim of the study, which should be stated at the end of the Introduction. Moreover, the authors should highlight the novelty and originality of their study and how it differs from other related studies in the literature.
Comments on the Quality of English LanguageMinor editing of English language required.
Author Response
Dear Reviewer,
Thank you for your comment. We addressed previous studies and emphasized the differences in detail in the discussion section:
“In comparison to previous studies, our investigation involved a substantially larger sample size, comprising 306 medical-records. To date, four studies have been published on this topic, with our research contributing significantly to understanding diagnostic markers and their role in the clinical management of acromegaly [2,3,13,17]. Unlike some earlier studies, our cohort encompassed three patient groups: naïve acromegaly, remission, and non-remission. The first of the cited studies, had the retrospective design and incorporated a total of 103 patients, providing valuable insights into post-surgical treatment outcomes [2]. However, it is noteworthy that the sample size was much smaller than our comprehensive study. One notable limitation of this study lies in the absence of a specific inclusion of a naïve acromegaly group. The second cited study, a retrospective analysis of 51 acromegaly patients, confirmed results on a smaller scale, without specifying a naïve acromegaly group[3]. In our study, we confirmed elevated phosphorus levels in the non-remission group and identified the highest phosphorus levels in patients with naïve acromegaly. Additionally, we confirmed a significant correlation between phosphorus levels and the SAGIT score. Notably, our study represents the largest real-life investigation to date on serum phosphorus and calcium as potential biomarkers for assessing disease status in acromegaly. In the third cited study, which included 73 acromegaly patients, participants were categorized into active and controlled acromegaly groups [13]. However, this study did not explore correlations with the SAGIT instrument. In our study, we expanded the analysis to 306 medical records and included three groups: naïve, non-remission, and remission. We observed the highest phosphorus and calcium levels in patients with naïve acromegaly. Importantly, we confirmed a correlation between serum phosphorus levels and elements of the SAGIT instrument, especially G and T, suggesting that phosphorus could be an effective biomarker in acromegaly diagnosis. Finally, in another study, encompassing 22 acromegaly patients and 22 with nonfunctioning pituitary adenomas, the focus was on patients assessed 3-6 months post-treatment [17]. In contrast to our study, acromegaly active group did not include division between a naïve and non-remission disease. In our research, we extended the follow-up period, particularly in the remission group, providing a more comprehensive perspective on treatment outcomes. Moreover, we observed serum Ca >9,89 mg/dL had the potency to discriminate between naïve and non-remission groups. Building upon prior findings, our study not only confirmed elevated serum calcium and phosphorus levels in the non-remission group but also revealed the highest calcium and phosphorus levels in patients diagnosed with naïve acromegaly. Furthermore, our investigation established a significant correlation between phosphorus levels and the SAGIT score, contributing to a more nuanced understanding of the biomarker's relevance in acromegaly.”
As suggested we also added the paragraph at the end of introduction:
“This study aims to assess the potential for the use of phosphorus and calcium as biomarkers in the clinical evaluation of patients with acromegaly and clarify their relationship with SAGIT and other common biomarkers (GH, IGF-1) in larger population size. It is essential to emphasize that previous studies in this field collectively constitute a limited dataset, none of which involved a patient cohort as large as ours. Only one study addressed correlations with SAGIT instrument [2]. As a novelty, we divided patients into three groups: naïve, non-remission and remission to accurately assess the changes in the various stages of the disease. To the date this is the largest real-life study investigating serum phosphorus and calcium as biomarkers of disease status in acromegaly”.
We hope that now it provides a clear rationale for the research question.
Reviewer 4 Report
Comments and Suggestions for Authors
As per my initial review, the work is potentially valuable. However, as I pointed out before there are many potential biases/confounding factors that have not been taken into account. I am unfortunately not happy with the inclusion, in response to my previous comments, of a line saying that osteopenia/osteoporosis, calcium supplements etc have been excluded. There is no mention in the Methods section as to how this has been achieved, nor has it been in the earlkier version. Therefore, unfortunately I maintain my intiial recommendation,
Author Response
Dear Reviewer,
Thank you for your remark. In material and methods we added the information that all patients underwent dual-energy X-ray absorptiometry to exclude osteopenia or osteoporosis, diagnosed according to the current guidelines. This is a routinely performed examination in patients with acromegaly in our center. A detailed medical history was taken, including medications and supplements that could affect Ca/ P concentrations. We also added this information in limitations section in the discussion:
“We applied strict inclusion criteria, and excluded all patients impaired renal function, thyroid dysfunction, osteopenia or osteoporosis, calcium supplementation, and primary hyperparathyroidism to avoid potential bias. Nevertheless, using a retrospective study design, it is possible that not all confounding factors were taken into account.”.
We hope you find our corrections well.